# Modeling Tool Use in Transformers via Computation Oracles

## Abstract

Prior literature has mapped the transformer architecture to classical models of computation (Strobl et al., 2024) and especially via circuit complexity, and analyzed how expressive power gets enhanced by adding computational resources like Chain-of-Thought (Merrill & Sabharwal, 2024), padding tokens (Merrill & Sabharwal, 2025a), and depth scaling (Merrill & Sabharwal, 2025b). Adding tools or function calls to transformer models has shown impressive empirical gains but their theoretical understanding remains under-studied. In this paper, we analyze function calls as *oracles* in classical complexity theory and provide a formal framework to analyze the expressive gains of augmenting transformer models with function calls and agentic cooperation. We show that (i) fixed-depth transformers with logarithmically many oracle calls can decide STCON via a weak state-transition oracle, (ii) repeated-block (unrolled) universal transformers are equivalent to bounded-bandwidth oracle computation when the oracle is block-realizable, and (iii) access to threshold decision oracles suffices to compute associated optimization objectives via binary search using only $O(\log B(n))$ adaptive queries, with between-query control logic implementable in constant-depth SMAT. Together, these results provide a complexity-theoretic framework for understanding tool use as an alternative mechanism for scaling test-time compute while clarifying when tool access can (and cannot) exceed the power of log-depth transformers.

## 1 Introduction and Related Work

A growing line of work characterizes transformer expressivity by mapping them to classical computation models (Strobl et al., 2024). Boolean circuits are natural due to parallelism; in particular, softmax-attention transformers (SMATs) with $O(\text{poly}(n))$ bits of precision are contained in DLOGTIME-uniform $\mathsf{TC}^0$ (Chiang, 2025), strengthening earlier simulations (Strobl, 2023; Merrill & Sabharwal, 2025c). This situates fixed-depth transformers as $\mathsf{SMAT} \subseteq \mathsf{TC}^0 \subseteq \mathsf{NC}^1$ where the known low-depth circuit hierarchy is:

$$\mathsf{NC}^0 \subsetneq \mathsf{AC}^0 \subsetneq \mathsf{TC}^0 \subseteq \mathsf{NC}^1.$$

Whether $\mathsf{TC}^0 = \mathsf{NC}^1$ remains open and if $\mathsf{TC}^0 \subsetneq \mathsf{NC}^1$, fixed-depth SMATs cannot capture even structurally simple sequential computations in $\mathsf{NC}^1$, such as uniform regular-language recognition.[1]

Expressivity can increase beyond fixed-depth by allowing dynamic test-time depth scaling (Giannou et al., 2023; Geiping et al., 2025; Yang et al., 2024). Merrill & Sabharwal (2025b) shows that log-depth unrolling in Universal Transformers (Dehghani et al., 2019) can recognize REG and decide STCON.[2] In practice, however, dynamic depth introduces training and serving complications: learning when to allocate compute can be unstable, and high-throughput inference favors fixed forward-pass structure (batching, kernel fusion).

In contrast, tool/function-call access has become standard in agentic systems (Yehudai et al., 2025; Malach et al., 2025). Yet its complexity-theoretic implications are comparatively underdeveloped.

---

[1] For example, deterministic finite automaton membership is in DLOGTIME-uniform $\mathsf{NC}^1$ (see, e.g., Arora & Barak (2009, Ch. 6)).

[2] $\mathrm{REG}_A = \{\langle w \rangle : A \text{ is a DFA and } A \text{ accepts } w\}$

We model tools as *oracles* and study the resulting relativized transformer classes, obtaining explicit equivalences and conditional separations, since many practical tool calls such as querying a knowledge base, executing a program, or checking satisfiability, naturally correspond to oracle access in the sense of classical complexity theory.

## 2    TOOL CALLS SOLVE GRAPH CONNECTIVITY

Under notations and definitions outlined in Section A.1, we consider the standard directed $s$-$t$ reachability problem:

$$\text{STCON} \ = \ \{(G, s, t) : G \text{ is a directed graph and } t \text{ is reachable from } s\}.$$

Since this problem is known to be complete for the class of logspace Turing machines (Reingold, 2008; Immerman, 1998) under common complexity conjectures, it cannot be solved accurately by fixed-depth transformers, which are upper bounded by the smaller class $\text{TC}^0$. Merrill & Sabharwal (2025b) show that a Universal Transformer (Dehghani et al., 2019) with uniform blocks repeating (unrolling) logarithmically many times can solve STCON. In this section we show that a fixed depth transformer with access to logarithmically many queries to a weak [3] oracle is capable of solving STCON.

### 2.1    A FIXED-DEPTH SMAT WITH STEP SOLVES STCON

As in Merrill & Sabharwal (2025b), we encode $G$ by its adjacency matrix $A \in \{0,1\}^{n \times n}$ together with $s, t \in [n]$ in unary, and use STEP as defined in Section A.5. For convenience, we may assume the input adjacency matrix already includes self-loops, i.e., $A[i,i] = 1$ for all $i$; this only changes the problem by an $\text{AC}^0$-preprocessing step.

**Lemma 1.** STCON $\in \text{SMAT}^{\text{STEP}}[\lceil \log_2 n \rceil, \Theta(n^2)]$. *In particular, there exists a fixed-depth SMAT controller that, using $\lceil \log_2 n \rceil$ adaptive queries to STEP (each of length $\Theta(n^2)$), decides whether $t$ is reachable from $s$.*

*Proof.* See Section A.6. We generalize this Lemma in Theorem 2 $\qquad\qquad\square$

### 2.2    EQUIVALENCE WITH LOG-DEPTH UNIVERSAL TRANSFORMERS

We relate the oracle-augmented model to log-depth (unrolled) universal transformers. Fix an input length parameter $n$.

**Universal transformers.**    Recall that an $(s, r, t)$-*universal transformer* consists of $s$ initial layers, followed by a block of $r$ layers repeated $d(n)$ times as a function of input length $n$, followed by $t$ final layers (where $s, r, t$ are independent of $n$).

**Definition 1** (State transition induced by a repeated block). *Fix $N, m, p$ and a repeated block $B$ consisting of $r$ transformer layers operating on $\mathcal{S}_n$. The block induces a (deterministic) state-transition function*

$$F_B : \mathcal{S}_n \to \mathcal{S}_n,$$

*mapping the residual stream immediately before $B$ to the residual stream after applying all $r$ layers of $B$.*

**Theorem 2** (Unrolling vs. oracle calls). *Let $U$ be an $(s, r, t)$-universal transformer unrolled $d(n)$ times on length-$n$ inputs, with repeated block $B$ and transition $F_B$. Then there exist (i) a fixed-depth SMAT controller $C$ and (ii) a function oracle $O_{F_B}$ such that*

$$C^{O_{F_B}}[d(n), \ell(n)]$$

*agrees with $U$ on all inputs of length $n$, where $\ell(n)$ is the number of bits used to encode a residual stream in $\mathcal{S}_n$.*

---

[3]The oracle is weak since it can be expressed by a fixed depth transformer. See Merrill & Sabharwal (2025b) Sec. D.

*Conversely, let $C$ be a fixed-depth SMAT controller that makes at most $d(n)$ adaptive queries to a function oracle $O$, and assume that $O$ is realizable by a fixed-depth block $B$ of $r = O(1)$ transformer layers operating on the same state representation $\mathcal{S}_n$. Then there exists an $(s', r', t')$-universal transformer $U'$ (for some constants $s', r', t'$ and $r' = O(1)$) unrolled $d(n)$ times that agrees with $C^O[d(n)]$ on all inputs of length $n$.*

*Proof.* See Section A.7. □

**Proposition 1** (Residual-stream encoding length). *Let $\mathcal{S}_n = (D_p^m)^N$ with $|D_p| = 2^p$, and let $\mathrm{Enc}_n : \mathcal{S}_n \to \{0,1\}^{\ell(n)}$ be any injective binary encoding of residual streams. Then*

$$\ell(n) \;\geq\; \log_2 |\mathcal{S}_n| \;=\; \log_2\big(|D_p|^{mN}\big) \;=\; mNp.$$

*In particular, when $N = \Theta(n)$ and $p(n) \geq c \log n$, it follows that $\ell(n) = \Omega(mn \log n)$.*

*Proof.* See Section A.8. □

## 3 FROM DECISION ORACLES TO OPTIMIZATION VIA BINARY SEARCH

### 3.1 OPTIMIZATION FROM THRESHOLD ORACLES

**Parameterized optimization problems.** Let $\mathcal{X}_n$ denote the set of instances of length $n$ under a fixed encoding. An *optimization function* is a mapping

$$\mathrm{OPT} : \bigcup_{n \geq 1} \mathcal{X}_n \to \mathbb{N}.$$

We assume a known polynomial bound on the optimum: there exists a polynomial $B(n)$ such that $\mathrm{OPT}(x) \in \{0, 1, \ldots, B(n)\}$ for all $x \in \mathcal{X}_n$.

Define the associated *threshold decision language*

$$\mathrm{THR_{OPT}} \;=\; \{(x, k) : x \in \mathcal{X}_n,\ 0 \leq k \leq B(n),\ \mathrm{OPT}(x) \geq k\}.$$

We also define the *bit-graph* language for the optimization output:

$$\mathrm{OPTBIT} \;=\; \{(x, i) : \text{the } i\text{-th bit of the binary representation of } \mathrm{OPT}(x) \text{ is } 1\},$$

where $i \in [\lceil \log_2(B(n) + 1) \rceil]$ and $\mathrm{OPT}(x)$ is written in $\lceil \log_2(B(n) + 1) \rceil$ bits.

**Arithmetic.** Let $k = \lceil \log_2(B(n) + 1) \rceil$. We represent a $k$-bit integer $x \in \{0,1\}^k$ by $k$ designated bit-cells in the residual stream (either $k$ token positions or $k$ fixed coordinates), each storing a bit in $\{0, 1\} \subseteq D_p$. We show that a constant-depth SMAT can implement the following operations on such encodings: (i) addition $x + y \pmod{2^k}$, (ii) comparison $x \geq y$, (iii) subtraction $x \mapsto x - 1$ on $x \in \{1, \ldots, 2^k - 1\}$ (and output 0 on $x = 0$), and (iv) division by 2, i.e. $x \mapsto \lfloor x/2 \rfloor$. Throughout, we use that $k = O(\log B(n))$ and that the precision regime satisfies $p(n) \geq c \log n$.

**Lemma 3** (Unbounded fan-in Boolean aggregation in constant-depth SMAT). *For any $k = O(\log B(n))$ and any subset $S \subseteq [k]$, there is a constant-depth SMAT module that, given bits $(b_i)_{i \in [k]}$ stored in designated bit-cells, computes $\bigvee_{i \in S} b_i$, $\bigwedge_{i \in S} b_i$, and $\mathrm{MAJ}_{i \in S}(b_i)$ and writes the result to a designated output cell.*

*Proof.* See Section A.9 □

**Lemma 4** (SMAT can implement one round of binary search). *There exists a constant depth $d = O(1)$ such that for all $n$ there is an $\mathrm{SMAT}_d$ computation which, on input $(L, U, b)$ where $L, U \in \{0, \ldots, B(n)\}$ are encoded in binary using $\lceil \log_2(B(n) + 1) \rceil$ bits and $b \in \{0, 1\}$, outputs $(L', U', M)$ where*

$$M = \left\lceil \frac{L + U}{2} \right\rceil, \quad (L', U') = \begin{cases} (M, U) & \text{if } b = 1, \\ (L, M - 1) & \text{if } b = 0. \end{cases}$$

*Proof.* Since operations can be composed via constant size transformer blocks, this immediately follows from Proposition 2

$\square$

**Theorem 5** (Optimization from decision via $O(\log B(n))$ oracle queries). *Let* OPT *be an optimization function with range contained in* $\{0, 1, \ldots, B(n)\}$ *on length-$n$ instances, and let* $\text{THR}_{\text{OPT}}$ *be its threshold decision language as above. Then*

$$\text{OPTBIT} \in \textsf{SMAT}^{\text{THR}_{\text{OPT}}}\Big[ \lceil \log_2(B(n) + 1) \rceil \Big].$$

*Equivalently, a fixed-depth* $\textsf{SMAT}$ *controller with access to the threshold decision oracle can compute* $\text{OPT}(x)$ *(in binary) using* $O(\log B(n))$ *adaptive oracle queries.*

*Proof.* See Section A.11 $\square$

### 3.2 SEPARATION BETWEEN TOOL USE AND log-DEPTH TRANSFORMERS

We now state the separation that motivates tool use. Let $\mathcal{U}_{\log}$ denote the class of languages decidable by log-depth (unrolled) universal transformers over the fixed residual-stream representation of Section 2.2. By Theorem 2, $\mathcal{U}_{\log}$ coincides with oracle-augmented computation for block-realizable oracles. If the oracle computes a language outside $\mathcal{U}_{\log}$, then oracle access strictly increases power.

The separation between $\textsf{TC}^0$ and $\textsf{NP}$ remains open but is likely to be true, and would be false if the circuit complexity hierarchy collapses to $\textsf{TC}^0$ and $\textsf{P} = \textsf{NP}$, hence, CLIQUE likely lies outside $\mathcal{U}_{\log}$ which is upper bounded by $\textsf{P}$

**Assumption 1** (Clique hardness for log-depth universal transformers). CLIQUE $\notin \mathcal{U}_{\log}$.

**Theorem 6** (Conditional strict separation via a powerful decision oracle). *Under Assumption 1,*

$$\mathcal{U}_{\log} \subsetneq \textsf{SMAT}^O\big[\lceil \log_2(n+1) \rceil, \ \ell(n)\big]$$

*for any $\ell(n)$ large enough to encode oracle queries of the form $(G, k)$, and for $O$ at least as powerful as* $\textsf{SMAT}$. *Moreover,* $\text{MAXCLIQUEBIT} \in \textsf{SMAT}^{\text{CLIQUE}}[\lceil \log_2(n+1) \rceil]$ *by Theorem 5.*

*Proof.* See Section A.12. $\square$

## 4 FUTURE WORK AND RESEARCH DIRECTIONS

We highlight several open directions under our formalism.

1. **Oracle Completeness under** $\textsf{SMAT}$**-Reductions.** Analogous to circuit classes characterized by complete problems under restricted reductions (e.g., $\textsf{AC}^0$-reductions), one may seek canonical problems that are complete for a given oracle $O$ under SMAT-reductions. Such a characterization would identify the precise class of problems solvable with $O$ and provide a structural taxonomy of tool capabilities.

2. **Adaptive vs. Non-Adaptive Querying.** Distinguish the power of adaptive and non-adaptive (parallel/batched) oracle access. These correspond to iterative tool use (e.g., Re-Act (Yao et al., 2023)) versus compiled parallel calls (e.g., LLMCompiler (Kim et al., 2024)). Formal separations or equivalences would clarify when adaptivity yields inherent computational gains, and when adaptivity can be traded off for higher number of queries that retain parallelism and thus reduce inference time.

3. **Agentic Cooperation:** $\textsf{SMAT}^{\textsf{SMAT}}$ Study layered constructions as models of multi-agent or tool-chaining systems. Does iteration strictly increase power, or collapse under closure?

4. **Imperfect Oracles.** Extend the framework to probabilistic or noisy oracles. Characterize robustness, error amplification, and the role of adaptivity under oracle uncertainty.

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

## A  APPENDIX

### A.1  NOTATION AND DEFINITIONS

### A.2  ORACLES

An oracle in computability and complexity theory is an abstract black box that can be queried to solve a specific subproblem in a single operation. Depending on the construction, an oracle may compute either a decision problem or a function problem. Formally, an oracle can be modeled as a language $A \subseteq \{0, 1\}^*$ (for decision queries) or as a function $f : \{0, 1\}^* \to \{0, 1\}^*$ (for functional queries). Standard background on oracle constructions can be found in texts such as Arora & Barak (2009).

### A.3  ORACLE MACHINES

An oracle machine is a computational model augmented with access to an oracle. Classical examples include oracle Turing machines, oracle Boolean circuits, and oracle RAM models. The machine is equipped with a special query mechanism that allows it to submit a string $x$ to the oracle and receive the answer in a single computational step. If the oracle represents a language $A$, the response is a single bit indicating whether $x \in A$; if the oracle represents a function $f$, the response is the value $f(x)$. Unless stated otherwise, we reduce function oracles to decision oracles via a standard bit-graph encoding (e.g., the predicate "the $i$th output bit of $g(q)$ equals 1").

Given a base complexity class $\mathcal{C}$ and an oracle $A$, the relativized class $\mathcal{C}^A$ consists of all problems solvable by machines of type $\mathcal{C}$ with oracle access to $A$ under the same resource bounds.

If the oracle is accessed multiple times during a computation, queries can be classified as *adaptive* or *non-adaptive*, depending on whether later queries may depend on earlier oracle answers.

### A.4  SMAT TRANSFORMERS

We adopt the formalization of transformer computation used by Merrill and Sabharwal (e.g., (Merrill & Sabharwal, 2024; 2025b)), specializing to *softmax-attention* (as in Chiang (2025)). Throughout, $\Sigma$ is a finite token alphabet and inputs are strings $x = x_1 \cdots x_n \in \Sigma^n$.

**Datatype and precision.**  Following Merrill & Sabharwal (2025b), we assume a numeric datatype $D_p$ whose elements are represented by $p = p(n)$ bits, together with implementations of the primitive operations appearing in the transformer computation graph (addition, multiplication, division, exp, and layer normalization). When we need exactness of numerical identities, we assume these primitives are $p$-precise in the sense of Merrill & Sabharwal (2025b) (Definition 4 therein), and that $p(n) = c \log n$ for a fixed constant $c > 0$.

**Residual stream.**  A (decoder-only) transformer maintains a residual stream $h_i^{(\ell)} \in D_p^m$ for each position $i \in [n]$ and layer index $\ell$. The base residual stream is

$$h_i^{(0)} \;=\; E(x_i) + \pi(i),$$

where $E : \Sigma \to D_p^m$ is an embedding map and $\pi : \mathbb{N} \to D_p^m$ is a positional encoding. As in Merrill & Sabharwal (2025b), we assume either a BOS token is present or that $\pi(1)$ is linearly separable from $\{\pi(i) : i \geq 2\}$, which suffices for standard routing constructions.

**Masked pre-norm.** Each sublayer is equipped with a learned mask $m \in D_p^m$ and receives, at each position, the normalized input

$$z_i \;=\; \mathrm{LN}(m \odot h_i),$$

where $\odot$ denotes coordinatewise multiplication and LN is layer normalization (Ba et al., 2016) (or RMS normalization (Zhang & Sennrich, 2019)). The sublayer produces an update $\delta_i \in D_p^m$ and the residual stream is updated as $h_i \leftarrow h_i + \delta_i$.

**Self-attention (softmax attention).** A multihead self-attention sublayer with $h$ heads is parameterized by an output projection $W \in D_p^{m \times m}$ and, for each head $k \in [h]$, matrices $Q_k, K_k, V_k \in D_p^{(m/h) \times m}$. Given the normalized inputs $z_1, \ldots, z_n$, the $k$th head computes, for each $i \in [n]$, queries $q_{i,k} = Q_k z_i$, keys $k_{j,k} = K_k z_j$, and values $v_{j,k} = V_k z_j$. For causal attention we restrict to $j \leq i$ (and for unmasked attention we allow all $j \in [n]$). The attention weights are

$$\alpha_{i,j,k} \;=\; \frac{\exp\left(\langle q_{i,k}, k_{j,k}\rangle\right)}{\sum_{t \leq i} \exp\left(\langle q_{i,k}, k_{t,k}\rangle\right)} \qquad \text{(causal)},$$

and the head output is $a_{i,k} = \sum_{j \leq i} \alpha_{i,j,k}\, v_{j,k}$. The sublayer output is

$$\delta_i \;=\; W \cdot \mathrm{concat}(a_{i,1}, \ldots, a_{i,h}) \in D_p^m.$$

(We use the standard softmax normalizer; when relevant, note that the *averaging-hard* ($\tau \to 0$) idealization in Merrill & Sabharwal (2025b) corresponds to a limiting high-temperature regime.)

**Feedforward sublayer.** A feedforward sublayer is parameterized by $W_1 \in D_p^{w \times m}$ and $W_2 \in D_p^{m \times w}$, and computes $\delta_i = W_2 \cdot \mathrm{ReLU}(W_1 z_i)$.

**Outputs and language recognition.** We assume a classifier head that reads from a designated position (e.g., the final position) and outputs a bit. Concretely, the head is a linear map $U : D_p^m \to D_p$ and the transformer outputs $\mathbf{1}[U(h_n^{(d)}) > 0]$. A transformer $T$ *recognizes* a language $L \subseteq \Sigma^*$ if $T(x) = 1$ iff $x \in L$ for all $x \in \Sigma^*$.

**Complexity classes of transformers.** Let $\mathsf{SMAT}_d$ denote the class of languages recognized by depth-$d$ softmax-attention transformers under the above model (with fixed $d$ independent of $n$), and let $\mathsf{SMAT} = \bigcup_{d=O(1)} \mathsf{SMAT}_d$. When we wish to emphasize the precision regime, we write $\mathsf{SMAT}_d[p]$.

**Oracle access and adaptivity.** Fix an oracle $O$. We write $\mathsf{SMAT}^O[t]$ for the class of languages decidable by an SMAT machine making at most $t = t(n)$ *adaptive* oracle queries, with SMAT computation between queries. Concretely, a computation in $\mathsf{SMAT}^O[t]$ proceeds as:

$$q_1 \leftarrow T_1(x), \quad a_1 \leftarrow O(q_1), \quad q_2 \leftarrow T_2(x, a_1), \;\ldots,\; q_t \leftarrow T_t(x, a_{<t}), \quad a_t \leftarrow O(q_t),$$

and outputs $T_{\mathrm{out}}(x, a_{\leq t})$, where each $T_i$ and $T_{\mathrm{out}}$ is in SMAT (or in $\mathsf{SMAT}_d$ for $\mathsf{SMAT}_d^O[t]$). We write $\mathsf{SMAT}_\|^O[t]$ for the *non-adaptive* (truth-table) variant in which all queries $q_1, \ldots, q_t$ are produced in one SMAT computation and the final output is an SMAT function of $(x, O(q_1), \ldots, O(q_t))$.

**Query length (interface bandwidth).** To model context window and tool schema limitations, we bound query length by a function $\ell(n)$. We write $\mathsf{SMAT}^O[t, \ell]$ (or $\mathsf{SMAT}_\|^O[t, \ell]$) to require that every oracle query $q$ satisfies $|q| \leq \ell(n)$ [4]

## A.5 The STEP oracle

We model a tool as a total function oracle. For each $n \in \mathbb{N}$, let $\mathrm{STEP}_n$ be the function

$$\mathrm{STEP}_n : \{0,1\}^{n \times n} \to \{0,1\}^{n \times n}, \qquad \mathrm{STEP}_n(R) \;=\; R \vee (R \circ R),$$

where $(R \circ R)[i, j] = \bigvee_{k=1}^{n} \left(R[i,k] \wedge R[k,j]\right)$ is Boolean relation composition and $\vee$ is entrywise OR. We write STEP for the family $(\mathrm{STEP}_n)_{n \geq 1}$. Operationally, the oracle takes as input an $n \times n$ Boolean matrix encoded in row-major order and returns the encoding of $\mathrm{STEP}_n(R)$.

---

[4]$\ell(n)$ is used to refer to both the length of the query, and amount of bits in the query, and the distinction is specified where necessary.

### A.6 PROOF OF LEMMA 1

*Proof.* Let $A \in \{0,1\}^{n \times n}$ be the (self-looped) adjacency matrix of $G$. Define a sequence of relations $(R_\ell)_{\ell \geq 0}$ by

$$R_0 := A, \qquad R_{\ell+1} := \text{STEP}_n(R_\ell) = R_\ell \vee (R_\ell \circ R_\ell).$$

The SMAT controller performs the following computation: (1) it places the bits of $R_0$ in a designated $n^2$-bit region of its state (e.g., the first $n^2$ token positions), (2) for $\ell = 0, 1, \ldots, L-1$ with $L = \lceil \log_2 n \rceil$, it queries the oracle on the current encoding of $R_\ell$ and overwrites the region with the returned encoding of $R_{\ell+1}$, and (3) it outputs the final bit $R_L[s,t]$.

Correctness follows from the standard doubling invariant used in the log-depth transformer construction of Merrill & Sabharwal (2025b). We claim that, for every $\ell \geq 0$ and $i, j \in [n]$,

$$R_\ell[i,j] = 1 \quad \Longleftrightarrow \quad G \text{ has a directed path from } i \text{ to } j \text{ of length at most } 2^\ell.$$

The base case $\ell = 0$ holds because $R_0 = A$ encodes edges (and self-loops). For the inductive step, $R_{\ell+1}[i,j] = 1$ iff either $R_\ell[i,j] = 1$ (a path of length $\leq 2^\ell$) or there exists $k$ with $R_\ell[i,k] = R_\ell[k,j] = 1$, yielding a concatenated path of length $\leq 2^\ell + 2^\ell = 2^{\ell+1}$. Conversely, any path of length at most $2^{\ell+1}$ can be split at its midpoint to produce such a $k$, so $(R_\ell \circ R_\ell)[i,j] = 1$. Thus the invariant holds.

Finally, any reachable pair $(s,t)$ has a simple path of length at most $n - 1 \leq 2^{\lceil \log_2 n \rceil}$. Hence $t$ is reachable from $s$ iff $R_L[s,t] = 1$, so the controller decides STCON. The controller depth is constant independent of $n$ (it only routes bits and reads out $R_L[s,t]$), and it makes exactly $L = \lceil \log_2 n \rceil$ oracle queries of length $\Theta(n^2)$[5]. $\qquad \square$

### A.7 PROOF FOR THEOREM 2

**State parameters.** Let

- $N = N(n) \in \mathbb{N}$ denote the *sequence length* (number of token positions) used by the model on length-$n$ inputs;

- $m = m(n) \in \mathbb{N}$ denote the *model width* (hidden dimension);

- $p = p(n) \in \mathbb{N}$ denote the *numeric precision* in bits;

- $D_p$ denote a *p-bit datatype*, i.e. a finite set of scalar values equipped with a fixed-length binary representation $\text{rep} : D_p \to \{0,1\}^p$ that is injective, and for convenience we take $|D_p| = 2^p$ (equivalently, rep is a bijection onto $\{0,1\}^p$).

A *residual stream* (model state) is an $N$-tuple of width-$m$ vectors over $D_p$, i.e.

$$H = (h_1, \ldots, h_N) \in (D_p^m)^N, \qquad \text{where each } h_i \in D_p^m. \tag{1}$$

We write $\mathcal{S}_n := (D_p^m)^N$ for the resulting state space at length $n$.

*Proof.* **First direction.** Fix a length-$n$ input $x$. Write the universal transformer computation as the composition

$$U = T_{\text{final}} \circ (F_B)^{d(n)} \circ T_{\text{init}},$$

where $T_{\text{init}}$ applies the $s$ initial layers, $F_B$ is the state transition induced by the repeated $r$-layer block $B$, and $T_{\text{final}}$ applies the $t$ final layers.

Let $\text{Enc}_n : \mathcal{S}_n \to \{0,1\}^{\ell(n)}$ be the canonical fixed-length encoding of residual streams induced by the componentwise representation $\text{rep} : D_p \to \{0,1\}^p$ and concatenation over coordinates and positions. Define the oracle $O_{F_B}$ by

$$O_{F_B}\big(\text{Enc}_n(H)\big) := \text{Enc}_n\big(F_B(H)\big) \qquad \text{for all } H \in \mathcal{S}_n.$$

---

[5]Here $\Theta$ denotes the length of the oracle query, while in the next section this denotes the *number of bits* required to encode this argument, which is $\Theta(n^2 \times width \times precision)$.

Consider the following controller computation: (i) compute $H^{(0)} := T_{\text{init}}(x)$ and form $q_0 := \text{Enc}_n(H^{(0)})$; (ii) for $\ell = 0, 1, \ldots, d(n) - 1$, query the oracle on $q_\ell$ to obtain $q_{\ell+1} = O_{F_B}(q_\ell)$; (iii) decode $H^{(d(n))} := \text{Enc}_n^{-1}(q_{d(n)})$, compute $T_{\text{final}}(H^{(d(n))})$, and output. By definition of $O_{F_B}$, we have $H^{(\ell+1)} = F_B(H^{(\ell)})$ for all $\ell$, hence the produced output equals $T_{\text{final}}\big((F_B)^{d(n)}(T_{\text{init}}(x))\big) = U(x)$. Since $s$ and $t$ are constants, all preprocessing/postprocessing is a fixed (constant-depth) computation and can be absorbed into a fixed-depth SMAT controller. Each query has length bounded by $\ell(n)$.

**Second direction.** Assume $C$ is fixed-depth and makes at most $d(n)$ adaptive queries to $O$ on length-$n$ inputs, and that $O$ is realized by a fixed transformer block $B$ on $\mathcal{S}_n$. Formally, there exists a fixed encoding $\text{Enc}_n$ of the state such that, for any query string $q$ that encodes a residual stream state $H$, the block $B$ maps $H$ to a new state whose encoding corresponds to $O(q)$ (i.e. $B$ implements the oracle function on the chosen representation).

We construct a universal transformer $U'$ that simulates the entire oracle-augmented computation by unrolling a constant-size repeated block. Because $C$ has fixed depth, its between-query computation defines a fixed state-update map on an augmented residual stream that stores (a) the controller's internal workspace, (b) the current oracle query region, and (c) the latest oracle answer region. These regions can be realized by reserving disjoint coordinate subsets of the width $m$ (and, if necessary, appending a constant number of auxiliary positions to the sequence, absorbed into $N$). Let $G$ denote the fixed-depth transformer block that updates the controller regions and writes the next oracle query into the query region. Let $B$ denote the fixed block implementing $O$ on the query/answer region. Define the repeated block of $U'$ to be the constant-depth composition $B \circ G$ (hence $r' = O(1)$ layers). Unrolling this block $d(n)$ times performs $d(n)$ oracle calls with the correct adaptive dependence, because $G$ computes the next query from the stored controller state and previous answers, and $B$ applies the oracle to that query. The initial layers of $U'$ write the initial controller state and first query; the final layers perform the readout. Therefore $U'$ agrees with $C^O[d(n)]$ on all length-$n$ inputs. $\qquad\square$

## A.8 PROOF FOR PROPOSITION 1

*Proof.* Injectivity of $\text{Enc}_n$ implies $2^{\ell(n)} \geq |\mathcal{S}_n|$. Since $|\mathcal{S}_n| = |(D_p^m)^N| = |D_p|^{mN}$ and $|D_p| = 2^p$, we obtain

$$\ell(n) \geq \log_2 |\mathcal{S}_n| = \log_2\big((2^p)^{mN}\big) = mNp.$$

The final asymptotic statement follows by substituting $N = N(n)$ and $p(n) \geq c \log n$. $\qquad\square$

## A.9 PROOF OF LEMMA 3

*Proof.* Fix a designated "collector" position. Using a single attention head, route from the collector uniformly over the indices in $S$ (i.e. ensure all keys in $S$ have equal score and all keys outside $S$ have negligible score); then the head output at the collector is the average $a = \frac{1}{|S|}\sum_{i \in S} b_i \in \{0, \frac{1}{|S|}, \ldots, 1\}$. A constant-size feedforward sublayer can threshold $a$ with margin at least $\frac{1}{2|S|}$ to recover: $\bigvee_{i \in S} b_i = \mathbf{1}[a \geq \frac{1}{2|S|}]$, $\bigwedge_{i \in S} b_i = \mathbf{1}[a \geq 1 - \frac{1}{2|S|}]$, and $\text{MAJ}_{i \in S}(b_i) = \mathbf{1}[a \geq \frac{1}{2}]$. Since $|S| \leq k = O(\log B(n))$ and $p(n) \geq c \log n$, these thresholds are representable and separable under the assumed $p$-bit datatype model, yielding exact Boolean outputs. The construction has constant depth independent of $n$. $\qquad\square$

## A.10 BASIC ARITHMETIC IN CONSTANT-DEPTH SMAT

**Proposition 2** (Basic arithmetic in constant-depth SMAT). *There exists a constant depth $d = O(1)$ such that for all $n$ there are $\text{SMAT}_d$ computations implementing:*

    *1.* $\text{SHR}_1(x) = \lfloor x/2 \rfloor$ *(right shift by one bit),*

    *2.* $\text{DEC}(x) = \max\{x - 1, 0\}$ *(subtract one with saturation at $0$),*

    *3.* $\text{GEQ}(x, y) = \mathbf{1}[x \geq y]$,

    *4.* $\text{ADD}(x, y) = (x + y) \bmod 2^k$,

*on $k$-bit inputs $x, y \in \{0, 1\}^k$ under the above encoding, where $k = \lceil \log_2(B(n) + 1) \rceil$.*

*Proof.* We use that constant-depth SMAT can (a) apply constant-arity Boolean functions to designated bit-cells via a constant-size feedforward sublayer, and (b) compute unbounded fan-in $\vee / \wedge$/MAJ over any selected subset of bit-cells by Lemma 3.

*Division by* 2. The map $\lfloor x/2 \rfloor$ is the right shift: $(\lfloor x/2 \rfloor)_i = x_{i+1}$ for $i \le k-2$ and $(\lfloor x/2 \rfloor)_{k-1} = 0$. This is implementable by a constant-depth routing module that copies each bit-cell to a fixed offset and writes 0 to the top cell.

*Subtract* 1. Let $x = (x_0, \ldots, x_{k-1})$ (least significant bit first). Define prefix-zero indicators $z_0 := 1$ and for $i \ge 1$, $z_i := \bigwedge_{j<i} \neg x_j$. Then for $x \ne 0$, $(x-1)_i = x_i \oplus z_i$ for all $i$, while for $x = 0$ we output 0. Each $z_i$ is an unbounded fan-in AND over a subset of bits, hence computable by Lemma 3, and $\oplus$ is constant-arity.

*Comparison.* Define $\mathrm{diff}_i := x_i \oplus y_i$ and $\mathrm{gt}_i := x_i \wedge \neg y_i$. Then

$$x > y \iff \bigvee_{i=k-1}^{0} \left( \mathrm{gt}_i \wedge \bigwedge_{j>i} \neg \mathrm{diff}_j \right), \qquad x = y \iff \bigwedge_{i=0}^{k-1} \neg \mathrm{diff}_i, \qquad x \ge y \iff (x > y) \vee (x = y).$$

This expression has constant Boolean depth with unbounded fan-in $\vee/\wedge$ and constant-arity gates, hence is computable by combining Lemma 3 with constant-arity feedforward logic.

*Addition.* It is classical that binary addition on $k$-bit integers is computable by DLOGTIME-uniform $\mathrm{TC}^0$ circuits of constant depth (e.g., via carry-lookahead constructions using threshold gates), and such circuits can be expressed using unbounded fan-in threshold/majority and constant-arity Boolean operations. By Lemma 3 (majority) together with constant-arity feedforward computation, the required constant-depth threshold-circuit computation can be realized within constant-depth SMAT on the designated bit-cells, yielding $\mathrm{ADD}(x, y)$. $\square$

In summary, constant-depth SMAT can implement the $O(\log B(n))$-bit arithmetic needed for binary search between oracle queries: right-shift (division by 2), decrement, comparison, and addition (for computing midpoints).

## A.11 PROOF FOR THEOREM 5

*Proof.* Fix $x \in \mathcal{X}_n$ and an output-bit index $i \in [\lceil \log_2(B(n)+1) \rceil]$. The procedure maintains bounds $L, U \in \{0, \ldots, B(n)\}$ satisfying the invariant

$$L \le \mathrm{OPT}(x) \le U.$$

Initialize $L \leftarrow 0$ and $U \leftarrow B(n)$. For rounds $r = 1, \ldots, T$ where $T = \lceil \log_2(B(n) + 1) \rceil$, do:

1. Compute $M = \lceil \frac{L+U}{2} \rceil$ using fixed-depth SMAT computation.

2. Query the oracle on $(x, M)$ to obtain the bit $b := \mathbf{1}[\mathrm{OPT}(x) \ge M]$.

3. Update $(L, U)$ to $(L', U')$ as in Lemma 4.

Finally, output the $i$-th bit of $L$.

Correctness follows from the standard binary-search argument. If $b = 1$ then $\mathrm{OPT}(x) \ge M$, so replacing $L$ by $M$ preserves the invariant; if $b = 0$ then $\mathrm{OPT}(x) \le M-1$, so replacing $U$ by $M-1$ preserves the invariant. Moreover, each update reduces the interval length by at least a factor of 2 (up to rounding), so after $T$ rounds we have $L = U = \mathrm{OPT}(x)$. The output is therefore the correct $i$-th bit.

Finally, the between-query computation in each round is implementable in constant-depth SMAT by Lemma 4, and there are exactly $T = O(\log B(n))$ oracle queries. $\square$

### A.12 PROOF FOR THEOREM 6

*Proof.* First, $\mathcal{U}_{\log} \subseteq \mathsf{SMAT}^{\mathrm{CLIQUE}}[\lceil\log_2(n+1)\rceil, \ell(n)]$ holds because the latter class contains fixed-depth $\mathsf{SMAT}$ computations with additional oracle access; in particular, the controller may ignore the oracle.

For strictness, observe that $\mathrm{CLIQUE} \in \mathsf{SMAT}^{\mathrm{CLIQUE}}[1, \ell(n)]$ by a single oracle query: on input $(G, k)$, query the oracle on $(G, k)$ and output the returned bit. If $\mathcal{U}_{\log}$ were equal to the oracle-augmented class, this would imply $\mathrm{CLIQUE} \in \mathcal{U}_{\log}$, contradicting Assumption 1. Hence the containment is strict.

Finally, Theorem 5 establishes $\mathrm{MAXCLIQUEBIT} \in \mathsf{SMAT}^{\mathrm{CLIQUE}}[\lceil\log_2(n+1)\rceil]$, demonstrating that $O(\log n)$ calls to a *decision* oracle suffice to compute the associated *optimization* value. $\qquad\square$

