# OpenReview forum: "Modeling Tool Use in Transformers via Computation Oracles"
_ICLR.cc/2026/Workshop/Sci4DL — Submitted to Sci4DL 2026_

### Official Review · Reviewer_8y4j · 2026-02-06

**Fit:** 2
**Significance:** 2
**Confidence:** 2

**Summary:**

This paper develops a complexity theory framework to study tool use by transformer models. They introduce SMAT^O with adaptive query budgets and query-length constraints and prove three main results: (i) a fixed-depth transformer can decide STCON with O(log n) calls to a simple STEP oracle that performs relation squaring-and-OR, (ii) repeated-block (unrolled) universal transformers are equivalent to fixed-depth controllers, and (iii) for parameterized optimization problems, access to the associated threshold decision oracle enables computing the optimum value using O(log B(n)) adaptive queries. The paper also gives a conditional separation showing that powerful decision oracles (e.g., CLIQUE) strictly increase power beyond log-depth unrolled transformers.

**Strengths:**

The theoretical results are consistent with known complexity-theoretic techniques (e.g., relation squaring, ..), and the assumptions are made explicit.
Their work helps explain why some multi-tool systems benefit from sequential reasoning while others do not.

**Suggestions:**

A toy example (e.g., graph reachability or simple optimization with bounded query length) showing how oracle bandwidth and adaptivity affect performance would greatly strengthen the paper’s relevance to practitioners. (I do acknowledge the tight 4 page limit, but would be something to add on the 5th page).
There is also a few smaller citation/formatting issues, that could be resolved rather quickly and polish the overall paper.

---

### Official Review · Reviewer_xKnF · 2026-02-27

**Fit:** 1
**Significance:** 2
**Confidence:** 2

**Summary:**

This paper uses tooling from computational complexity to study the expressiveness of transformers when "tool calls" are allowed.
In particular, a fixed-depth transformer is allowed logarithmically many calls to an oracle, also modeled as a fixed-depth transformer.
The authors show that this model: (1) can decide STCON in logarithmically many oracle calls, (2) is equivalently expressive to an existing architecture where blocks may be unrolled logarithmically many times, and (3) can also decide binary search given access to a threshold oracle.

**Strengths:**

S1. To my knowledge, this paper is the first to investigate transformer expressivity given tool calls.

S2. The theoretical results are non-trivial.

**Suggestions:**

W1. This manuscript is confusing to read and not well-written.

W2. While these results and techniques would be of interest to researchers in computational complexity, it is not clear to me how a practitioner might use them.

W3. This paper does not analyze trained models with hypotheses and controlled experiments, which does not align well with the scope of this workshop.

---

### Meta-Review · Area_Chair_ar4N · 2026-03-01

**Recommendation:** Reject

**Metareview:**

Recommending rejection, due to lack of fit for the workshop.

---

### Decision · Program_Chairs · 2026-03-02

Reject